# The Assessment of a Novel Endoscopic Ultrasound-Compatible Cryocatheter to Ablate Pancreatic Cancer

**DOI:** 10.3390/biomedicines12030507

**Published:** 2024-02-23

**Authors:** John M. Baust, Anthony Robilotto, Isaac Raijman, Kimberly L. Santucci, Robert G. Van Buskirk, John G. Baust, Kristi K. Snyder

**Affiliations:** 1CPSI Biotech, Owego, NY 13827, USA; 2Phase Therapeutics, Inc., Owego, NY 13827, USA; 3Department of Medicine-Gastroenterology, Baylor College of Medicine, Houston, TX 77030, USA; 4GI Alliance, Houston, TX 77030, USA; 5Center for Translational Stem Cell and Tissue Engineering, Binghamton University, Binghamton, NY 13902, USA; 6Department of Biological Sciences, Binghamton University, Binghamton, NY 13902, USA

**Keywords:** cryoablation, *FrostBite*, pressurized sub-cooled nitrogen, tissue engineered models, porcine model, pancreatic cancer, liver cancer, endoscopic ultrasound, endoscopy, NOTES

## Abstract

Pancreatic ductal adenocarcinoma (PDAC) is a highly lethal disease that may be treated utilizing thermal therapies. Cryoablation is an effective, minimally invasive therapy that has been utilized for the treatment of various cancers, offering patients a quicker recovery and reduced side effects. Cryoablation has been utilized on a limited basis for the treatment of PDAC. With the recent reports on the success of cryoablation, there is a growing interest in the use of cryoablation as a standalone, minimally invasive procedure to treat PDAC. While offering a promising path, the application of cryoablation to PDAC is limited by current technologies. As such, there is a need for the development of new devices to support advanced treatment strategies for PDAC. To this end, this study investigated the performance of a new endoscopic ultrasound-compatible cryoablation catheter technology, *FrostBite*. We hypothesized that *FrostBite* would enable the rapid, effective, minimally invasive delivery of ultra-cold temperatures to target tissues, resulting in effective ablation via an endoscopic approach. Thermal properties and ablative efficacy were evaluated using a heat-loaded gel model, tissue-engineered models (TEMs), and an initial in vivo porcine study. Freeze protocols evaluated included single and repeat 3 and 5 min applications. Isotherm assessment revealed the generation of a 2.2 cm diameter frozen mass with the −20 °C isotherm reaching a diameter of 1.5 cm following a single 5 min freeze. TEM studies revealed the achievement of temperatures ≤ −20 °C at a diameter of 1.9 cm after a 5 min freeze. Fluorescent imaging conducted 24 h post-thaw demonstrated a uniformly shaped ellipsoidal ablative zone with a midline diameter of 2.5 cm, resulting in a total ablative volume of 6.9 cm^3^ after a single 5 min freeze. In vivo findings consistently demonstrated the generation of ablative areas measuring 2.03 cm × 3.2 cm. These studies demonstrate the potential of the *FrostBite* cryocatheter as an endoscopic ultrasound-based treatment option. The data suggest that *FrostBite* may provide for the rapid, effective, controllable freezing of cancerous pancreatic and liver tissues. This ablative power also offers the potential of improved safety margins via the minimally invasive nature of an endoscopic ultrasound-based approach or natural orifice transluminal endoscopic surgery (NOTES)-based approach. The results of this pre-clinical feasibility study show promise, affirming the need for further investigation into the potential of the *FrostBite* cryocatheter as an advanced, minimally invasive cryoablative technology.

## 1. Introduction

To address pancreatic cancer, it is evident that novel therapeutic approaches are needed. The conventional strategies of surgical resection or chemotherapy followed by chemo-radiation therapy yield a median survival of six to twelve months [1,2,3]. However, nearly half of patients with no metastases face challenges in tumor resection due to factors such as vascular invasion, compromised general health, or the absence of suitable surgical techniques [4,5,6,7]. For unresectable PDAC, available treatments are predominantly palliative, encompassing chemotherapy options like gemcitabine, oxaliplatin, cisplatin, 5-FU, and taxanes [8,9,10,11,12,13,14]. Locally advanced PDAC is typically managed with FOLFIRINOX or Gem/Abraxane with or without radiation, notwithstanding the associated significant complications [5,6,7,15,16,17,18,19,20]. Investigations into the use of thermal ablation, including radiofrequency ablation (RFA), microwave therapy, high-intensity focused ultrasound (HIFU), and cryoablation (CA), are currently underway as potential interventions for unresectable PDAC [8]. With the annual global incidence surpassing 490,000 cases and a 5-year survival rate of <10%, resulting in a projected 460,000 deaths in 2023, there is a pressing need for innovative approaches to treat PDAC, especially considering that the survival rate has remained unchanged for over two decades despite extensive research [1,2,3,21,22,23].

In pursuit of this objective, cryoablation has demonstrated effectiveness as a minimally invasive treatment option for various cancers, experiencing significant growth in recent years [24,25,26]. Cryotherapy stands out among other treatments, particularly hyperthermic (heat-based) ablative approaches, offering advantages such as shorter procedure times, increased safety, reduced pain, diminished operator stress, and fewer unintended side effects [27,28,29,30,31,32,33,34,35,36]. Cryoablation has been shown to elicit a systemic immune response, expanding its impact beyond focal destruction within the frozen tissue mass to potentially impacting microsatellite, multifocal, and metastatic disease [37,38,39,40,41,42]. Visualization of cryolesion formation using ultrasound is possible. Importantly, cryolesions do not exhibit post-treatment growth, distinguishing it from hyperthermic ablation, where lesions may enlarge for several days, leading to a heightened risk of complications [43,44,45]. Additional reported benefits of cryoablation include enhanced depth of penetration, the ability to be used in conjunction with other treatments in an adjunctive role, improved hemostasis, and the ability to target unresectable tumors [46,47,48]. The delivery of cryoablative doses to tissues not conventionally targeted by cryoablation has expanded through the use of percutaneous needles or catheter-based approaches [49,50,51,52,53,54,55,56,57,58,59]. However, for these therapies to achieve clinical viability, the cryotherapeutic approach must effectively deliver ablative temperatures in a confined, controlled, and time-efficient manner.

The application of cryotherapy as a minimally invasive primary or combined treatment option for PDAC has been previously reported [42,60,61,62,63,64,65,66]. Despite promising results, its utilization remains largely investigational. Cryoablation achieves the objective of reducing tumor mass while being less invasive than traditional surgery. A percutaneous approach using a cryoprobe can generate multiple cryolesions in a single procedure. Targeting PDAC tumors via a percutaneous approach can be challenging due to the pancreas’s location. Recently, our research group has developed an advanced cryoablation system and an endoscopic ultrasound (EUS)-compatible cryocatheter, significantly expanding the potential use of cryoablation in the treatment of PDAC.

In this investigation, we evaluated the performance of a novel endoscopic-compatible cryocatheter, *FrostBite*, in combination with the pressurized sub-cooled nitrogen (PSN) cryoconsole (Figure 1), for potential application for the targeted ablation of pancreatic tissue, including PDAC. Employing a mixed-phase liquid (LN2) and gaseous (N2) nitrogen cryogen, the PSN device delivers sub-lethal temperatures to a targeted tissue via unique cryoapplicators specifically designed for ablating various tissues [48,67]. This study specifically focused on assessing the ability of *FrostBite* to rapidly and effectively deliver an ablative dose for the ablation of PDAC and other gastrointestinal cancers. Employing acellular hydrogels in vitro, TEMs, and a pilot in vivo porcine study, our objective was to evaluate *FrostBite*, demonstrating its potential use for the targeted ablation of unwanted pancreatic tissue, including PDAC.

## 2. Materials and Methods

### 2.1. FrostBite and the Pressurized Sub-Cooled Nitrogen (PSN) System

All evaluations were conducted using the PSN cryosurgical device (CPSI Biotech, Owego, NY, USA) set to an input N_2_ pressure of 1500 psi. The *FrostBite* EUS cryocatheter (Phase Tx., Owego, NY, USA) employed in these studies was an 8 Fr (2.6 mm OD) × ~60 cm long flexible cryocatheter. The catheter had a 1.3 mm (diameter) × 13 cm (length) nitinol needle tip with a distal 3 cm ablation zone and was connected to the PSN system via a ~3 m umbilical (Figure 1).

### 2.2. Isotherm and Calorimetry Testing

Isotherm Testing: The distal ablation tip of *FrostBite* was introduced into an isothermal test fixture comprising an acrylic box filled with ultrasound gel maintained at 35 °C (±2) [67]. A thermocouple mandrel, equipped with 5, 36 Ga type-T thermocouples, was inserted into the fixture. The midpoint of the cryocatheter freeze segment was positioned on the mandrel, ensuring that the thermocouples were radially spaced at 1, 5, 7.5, 10, and 15 mm from the cryocatheter surface. With the cryoprobe centered in the test fixture, a single and repeat (double) 3 and 5 min freeze cycle was executed. Temperatures at the various points on the mandrel were recorded at a rate of 1/sec using an Omega OMB-DaqScan 2000 Series device (Omega Engineering, Norwalk, CT, USA).

Calorimetry Testing: Calorimetry testing involved placing a 30 mm t-style magnetic stir-bar at the bottom of a 20 oz double-walled foam-insulated vessel filled with 454 mL of water at 32 °C ± 1 °C [67]. The vessel had a ~20 mm thick polystyrene lid containing two tight-fitting holes. The *FrostBite* cryoneedle was inserted through one hole, and the distal end of the cryoprobe was positioned 20 mm from the bottom of the vessel to submerge the entire freeze zone without contacting the stir bar or vessel wall. A 21 ga (0.032 inch) type-T thermocouple needle was inserted through the second hole and into the water to monitor temperature changes. The setup was then positioned on a magnetic stir plate set to the highest spinning setting, ensuring uniform rotation (setting 8 out of 11). A 5 min freeze was conducted, and water temperatures were recorded at T = 0 (start) and subsequently at 30 s intervals from T = 3 min to T = 5 min. The total wattage at each time point was calculated and converted to the average W/mm^2^ of surface area.

### 2.3. Cell and 3D TEM Culture

Cells were cultured in T-75 CellTreat flasks (CellTreat, Shirley, MA, USA) within 95% O_2_/5% CO_2_ incubators and were passaged upon reaching 80–85% confluence. Media replacement was performed every three days. PANC-1 cells (ATCC, Rockville, MD, USA) were cultivated in DMEM (Caisson DML10) supplemented with 10% FBS (Peak Serum, PS-FB3, Wellington, CO, USA) and 1% penicillin/streptomycin (Lonza, 17-602E, Walkersville, MD, USA). All experiments were conducted on cultures between passages 5 and 20. Note: the PANC-1 cell line is not listed on the International Cell Line Authentication Committee’s database of commonly misidentified cell lines (Version 12, 16 January 2023).

To generate TEMs, a solution of rat tail type I collagen (BD Bioscience, Bedford, MA, USA) was utilized to create 0.2% *w/v* gel matrices following standard operating procedures (SOP). Cells (0.75–1 × 10^6^ cells/mL) were suspended in the collagen solution and dispersed into 3 mm × 40 mm TEM ring fixtures in 100 mm Petri dishes. Constructs were allowed to solidify for 30 min in a 37 °C hybridization oven as per Robilotto et al. and Baust et al. [68,69]. After solidification, 15 mL of cell culture medium was added to cover the TEMs, and the dishes were placed in the incubator. TEMs were then cultured for 24 h before use in experiments.

### 2.4. Freeze Procedure

Experiments were conducted within a laminar flow hood to maintain sample sterility. To create the 3D configuration, individual cell-seeded TEMs were stacked, as per [67], and submerged in warm circulating culture medium within an acrylic box. The box was positioned on a heat pad and stir table, and the *FrostBite* cryoneedle and a thermocouple array were inserted into the TEM. TEMs were held in the bath until the temperatures reached equilibrium at 32 °C (±2). A single 5 min freeze was conducted. Temperatures within the TEM and the bath were monitored throughout the process. The multipoint thermocouple array enabled monitoring of temperatures within the TEM at midpoint of the freeze zone at distances of 7.5 mm, 10.5 mm, 13 mm, and 16 mm radially from the cryoprobe surface. An Omega TempScan collected temperature data at 1 s intervals throughout the freeze cycle. Following freezing, TEMs were allowed to thaw for 30 min in the warm circulating bath and then disassembled. Individual TEM layers were then placed in culture for recovery and subsequent assessment.

### 2.5. Cell Viability Assessment

After thawing, TEM layers were assessed using calipers to determine the iceball diameter resulting from freezing with *FrostBite*. Iceball radii were measured at cardinal locations around the probe to evaluate the symmetry of the freeze zone. TEMs were placed into culture, and viability was assessed at 24 h and 72 h post-freeze. In situ sample viability was evaluated using the Calcein-AM and propidium iodide live/dead assay (Cal/PI; Molecular Probes, Eugene, OR, USA). Briefly, culture medium was removed, and a working solution of 5 µg/mL Calcein-AM (indicating live cells) and 4 µg/mL propidium iodide (indicating necrotic cells) in 1X PBS (Corning Life Sciences, Corning, NY, USA) was applied directly to each sample. The samples were then incubated in the dark at 37 °C for 60 min (±1 min). Fluorescent staining was assessed using a Zeiss Axio Observer 7 with ZEN 2.3 Pro software (Carl Zeiss AG, Oberkochen, Germany). Panoramic digital images covering the midline of the freeze zone were generated by stitching together a 6 × 30 set of overlapping images using a 10X objective. A 2 mm scale bar was added to each image to facilitate direct comparison. The Zeiss ZEN 2.3 Pro software measurement tool was then used to measure the diameters of the necrotic zones within the TEMs.

### 2.6. Pilot In Vivo Porcine Study

In vivo studies were conducted using a female Yorkshire pig model at University Hospital animal lab in the Cleveland Medical Center (Cleveland, OH, USA) under IACUC approval. For this study, the animal was sedated, and an Olympus GF-UCT140P-AL5 endoscopic ultrasound (EUS) (Olympus Corporation, Center Valley, PA, USA) was inserted down the throat and into the stomach of the pig. Once in position, with the tip of the EUS scope in a straight and unlocked position, *FrostBite* was inserted through the working channel, and the *FrostBite* handle was affixed to the EUS scope handle via the twist connector. The EUS scope was then maneuvered within the stomach to visualize target tissue within the pig liver and pancreas using endoscopic ultrasound. The liver was used as a surrogate tissue in this pilot study, given the pig pancreas is very small and targeting and creating multiple ablations is not possible. Once identified, using ultrasound guidance, the *FrostBite* cryoneedle was advanced into the target, and a 3 or 5 min single freeze procedure was applied. A double 3 min (3/3/3) freeze procedure was also applied. Probe temperature, freeze time, and lesion location and size were recorded. Real-time visualization of lesion formation was accomplished using endoscopic ultrasound. Following freezing, the tissue was allowed to passively thaw, and the needle was repositioned for another freeze. Upon procedure completion, *FrostBite* and the endoscope were removed, the animal sacrificed, and the tissue excised for gross pathological assessment. In total, eight unique ablation sites were created and assessed to evaluate in situ performance. Quantitative assessments were performed to assess the size (diameter and length) of each lesion.

### 2.7. Data Analysis

All isotherm and TEM studies had a minimum of 3 repeats. The pilot porcine study consisted of 8 replicate freezes. The data were combined and averaged (±standard deviation (SD)) to ascertain the mean iceball size, isotherm distribution, cooling power, and ablative diameter. For calorimetry studies, total wattage was calculated based on temperature change at each time point and converted to average W/mm^2^ of surface area. Total wattage and W/mm^2^ calculations between 3 and 5 min were averaged (±SD). Single-factor ANOVA was used to determine statistical significance where noted.

## 3. Results

### 3.1. Isotherm Distribution

To evaluate the performance of *FrostBite*, studies analyzing isotherm distribution around the ablation segment were undertaken. Single and repeat 3 and 5 min freeze protocols were selected as previous reports have shown that extended freeze times are not required [48]. After a single 3 min freeze in 35 °C (±2) ultrasound gel, *FrostBite* generated an iceball measuring 17.1 mm (± 0.2) in diameter and 32.4 mm (±1.0) in length (Table 1). Extension to 5 min resulted in a 21.7 mm (±0.8) diameter by 35.1 mm (±0.7) long iceball. Similarly, a double 3 min freeze with a 5 min intermediate passive thaw (3/5/3) yielded a 20.6 mm (±0.5) by 34.2 mm (±0.3) iceball, whereas a double 5 min freeze (5/5/5) yielded a 27.5 mm (±0.6) by 38.2 mm (±0.7) iceball (Table 1).

The absolute size of the iceball, though informative, does not provide insights into the distribution of isotherms within the frozen mass. A more clinically significant metric involves the understanding of the spread of critical isotherms or the minimal lethal temperature, the temperature threshold at which complete cell destruction occurs in the targeted tissue. Previous in vitro investigations have reported a minimal lethal temperature of −25 °C for PANC-1 cells following a single 5 min freeze exposure and −20 °C for a double 5 min freeze exposure [68,70,71]. These and various other investigations have established that exposure to the minimal lethal temperature for ≥30 s can yield cell death. Consequently, shorter freeze intervals are feasible when the target temperatures are reached. To assess the temperature distributions within the frozen mass produced by *FrostBite*, an array of thermocouples was employed (Figure 2). Data analysis revealed diameters of 11.6 mm (±0.5) for the −20 °C isotherm and 9.6 mm (±0.3) for the −30 °C isotherm after a single 3 min freeze, signifying an ablative volume >1.5 cm^3^ (Table 2). The application of a 5 min freeze increased the −20 °C and −30 °C isotherm penetration to 14.8 mm (±0.4) and 12.1 mm (±0.4), respectively, equating to an ablative volume of >2.5 cm^3^. The application of a double 3 min freeze protocol yielded similar results to the single 5 min freeze with the −20 °C and −30 °C isotherms at 14.0 mm (±0.6) and 11.6 mm (±0.5), respectively, equating to an ablative volume of >2.45 cm^3^. A double 5 min freeze yielded −20 °C and −30 °C isotherm penetration to 18.6 mm (±0.6) and 15.0 mm (±0.6), respectively, equating to an ablative volume of >4.4 cm^3^ (Table 2).

### 3.2. Calorimetry Testing

The power of the ablation segment (wattage) is often employed to assess the efficacy of hyperthermal (heat-based) devices. While less common in cryoablation devices, a series of calorimetry studies were conducted to characterize the heat extraction (ablation) power of *FrostBite*. A starting water temperature of 32 °C (±1) was selected as previous reports have shown minimal ice formation (<1 mm) on the probe surface at this temperature in the calorimetry vessel [67,72]. No ice was observed at warmer temperatures, suggesting inefficient flow (complete boiling) of the cryogen. Conversely, at colder temperatures, the ice started to accumulate along the freeze segment of the probe, insulating the freeze surface from the surrounding water. As such, calorimetric analysis was performed with a starting water temperature of 32 °C (±1) (Table 3). Assessment of water temperature revealed an average drop of 4.9 °C (±0.8) and 8.4 °C (±1.2) following a 3 or 5 min freeze, respectively. This is calculated to an average cooling power of 52.1 W (±5.7) or an average cooling power of 1.1 W/mm^2^ (±0.03) of surface area for freeze intervals between 3 and 5 min (Table 3).

### 3.3. Tissue Engineered Model Freezing

Although the use of acellular hydrogels and calorimetric testing offers valuable insights into the formation and spread of critical isotherms and the cooling power of cryoprobes, such tests do not offer insight into the cellular or tissue response to the freezing process. Utilization of in vitro 3-dimensional TEMs in a clinically analogous test setup provides a method to evaluate both the cryocatheter performance and cellular response. TEMs facilitate accelerated research and enable subsequent recovery studies to evaluate the effects of delayed tissue damage [69]. This model has been shown to reduce the expense and burden of exploratory animal studies while providing insight into in vivo response [69,72]. The PANC-1 cell line was chosen for these studies.

TEM Viability: The ablative zone created by the *FrostBite* EUS cryocatheter in TEMs was assessed using fluorescence microscopy post-thaw. Additionally, size and isotherm distribution were also assessed. To determine the extent of cell death within the frozen volume, at 24 h and 72 h following a 5 min freeze, replicate TEMs were stained with Calcein-AM (green, live) and propidium iodide (red, necrotic) (Figure 3). Replicate sister samples were utilized to assess viability following 72 h of recovery as Calcein-AM and propidium iodide are single end-point assays. Non-frozen controls revealed minimal cell death over the 72 h of culture. Iceball diameter and the extent of cell death in TEMs were measured using the ZEN software, and results were averaged from triplicate experiments (Table 4).

The freeze diameter averaged 25.1 mm (±0.39) (volume = 13.0 cm^3^) for the single 5 min freeze. At 24 h post-thaw, the PANC-1 TEMs revealed a 20.1 mm (±2.1) necrotic zone diameter, yielding a necrotic volume of 6.9 cm^3^. At 72 h post-thaw, the necrotic diameter was similar to that at 24 h, with a diameter of 20.6 mm (±1.4) and a necrotic volume of 6.8 cm^3^ (*p* = 0.67; 24 h vs. 72 h, respectively). Previous studies have shown that necrotic zones created with cryoablation remain similar in size during the recovery interval versus heat-based ablation, wherein lesions typically increase in size over several days to weeks post-ablation [73,74]. The necrotic zone of a cryolesion has been reported to either maintain or decrease during recovery in the TEM model. This has been attributed to a combination of cellular recovery, infiltration, and regrowth in the periphery of the cryogenic lesion over the 3-day recovery period in the region where cells only experience a thermal insult warmer than the minimal critical temperature [67,69,72].

### 3.4. In Vivo Porcine Study

A pilot in vivo porcine study was then conducted to further assess *FrostBite* EUS cryocatheter performance. In this study, eight lesions were created in the liver using an endoscopic ultrasound approach. The porcine liver was used as a surrogate tissue as the porcine pancreas is very small, and it is difficult to perform multiple freezes. The Olympus EUS endoscope was positioned within the stomach. With the EUS scope in an unlocked position, the 8 Fr *FrostBite* EUS cryocatheter was inserted down the working channel and secured to the EUS scope connector. Once *FrostBite* was inserted, the distal end of the EUS scope was positioned against the stomach wall, and the liver was visualized under ultrasound. Once a target area was identified via ultrasound, the cryoablation needle tip of *FrostBite* was advanced from the EUS scope, through the stomach wall and into the target. Cryoneedle insertion was visualized via ultrasound (Figure 4). Once in position, the cryoablation procedure was initiated. Freezing of the tissue surrounding the cryoneedle was able to be visualized using EUS in 30 s (±5), and growth (size) of the cryolesion was monitored throughout the freeze procedure in real time via EUS imaging (Figure 4A). Following the freeze procedure, the tissue was allowed to passively thaw, and the cryoneedle was withdrawn back into the working channel, the EUS scope was repositioned, the cryoneedle was redeployed, and another freeze procedure was conducted. On average, the cryoneedle was found to release from the tissue (passively thaw) within 65 s (±15 s) following the termination of cryogen flow. At ~10 s before completion of each freeze procedure, lesion diameter was measured using EUS imaging (Figure 4). The application of a single 3 min freeze yielded an average lesion diameter of 2.1 cm (±0.05), whereas a repeat 3 min freeze yielded a 2.4 cm diameter frozen mass (Table 5 and Figure 4). In total, six single 3 min, one single 5 min, and one repeat 3 min (3/3/3) freeze protocols were applied. For one 3 min freeze, the cryoneedle was intentionally placed in close proximity to a blood vessel (~0.6 cm by EUS measurement) to assess the impact of freezing on blood vessel integrity (Figure 5). Vascular flow was monitored via EUS Doppler imaging throughout the freeze to assess the impact on blood flow (Figure 5). In this case, there was no apparent impact on blood flow during freezing, and the iceball (frozen tissue) appeared to sculpt around the blood vessel during the procedure (Figure 5).

Following the freeze procedures, the animal was sacrificed, and an analysis of the lesions was performed (Figure 6). A gross pathological assessment revealed the creation of a 2.03 cm (±0.06) diameter × 3.15 cm (±0.13) cryolesion on average following a single 3 min freeze (Table 5). Extension of the freeze time to 5 min resulted in an increase in lesion size to 2.2 cm × 3.4 cm. Similarly, the application of a repeat 3 min freeze (3/3/3) resulted in a 2.5 cm × 3.5 cm lesion (Table 5 and Figure 6B). Pathological examination of the 3 min freeze that was conducted in close proximity to a blood vessel revealed that the frozen tissue did, in fact, sculpt around the blood vessel as was observed under ultrasound, and there was no obvious damage to the blood vessel wall (Figure 6D).

## 4. Discussion

While the use of percutaneous cryoablation to target PDAC has been reported, due to anatomical limitations, targeting PDAC using a percutaneous approach is challenging [4,5]. The potential benefit of targeting PDAC using an EUS approach has been reported using RFA-based ablation catheters. These studies suggest that using an EUS approach increases access to pancreatic lesions that can be obtained using a minimally invasive procedure [75,76,77,78,79,80,81,82]. Development of EUS-based cryotechnologies has been limited due to cryoprobe size and cryogen power limitations. Leveraging an innovative mixed-phase nitrogen cryogen source (PSN), *FrostBite* demonstrated the ability to quickly administer sub-lethal temperatures through microcatheter tubing, allowing for the efficient and repeatable ablation of tissue in situ via an EUS approach (Figure 1). The drive to offer patients and healthcare providers more sophisticated and minimally invasive ablative therapies has spurred the creation of the PSN cryosurgical system. In contrast to the majority of commercially available cryoablative devices, which depend on the Joule–Thomson expansion of a gas (e.g., Argon), the PSN system utilizes a mixed-phase nitrogen cryogen. The integration of liquid and gaseous nitrogen enables the swift establishment of cryogen flow, the delivery of ultra-cold nadir temperatures, and the high thermal conductivity characteristic of liquid cryogens [48]. The PSN system’s compatibility with extended lengths of hypodermic tubing creates a versatile platform compatible with a diversity of cryoprobe configurations. In this study, we assessed the prototype *FrostBite* EUS cryocatheter for transesophageal endoscopic ultrasound-guided ablation of pancreatic and liver tissue.

Calorimetric analysis revealed that *FrostBite* provides an average cooling power of 52.1 W, equivalent to 1.1 W/mm^2^. Cooling power (wattage) is not a standard metric in cryoablation probe reporting. The available data, however, suggest that existing argon and nitrous oxide cryoprobes typically deliver cooling power in the range of 0.1–0.25 W/mm^2^ [67,69]. As such, compared to current cryodevices, *FrostBite* is significantly more powerful (an estimated 4–10 fold increase) [48,72,83,84].

Benchtop assessment of the *FrostBite* performance revealed the attainment of sub-freezing temperatures within 20 s and a surface temperature of <−90 °C within 60 s (Figure 2). Critical isotherm assessment revealed the −20 °C isotherm penetrated to a depth of 11.6 mm (±0.5) and 14.8 mm (±0.4) following a single 3 and 5 min freeze, respectively (Table 1). In comparison, the −20 °C isotherm for JT-based cryosystems is reported to penetrate < 6 mm and ~10 mm following 5 and 15 min of freezing, respectively [69,83,84]. Clinically, the position of the −20 °C and −40 °C isotherms are often important reference points [31,32,46,53]. With −20 °C to −25 °C reported as the minimal lethal temperature range for PDAC, we focused on the −20°C isotherm as a potential thermal target for PDAC ablation [68,70,71]. To this end, in comparison with current cryosystems, *FrostBite* and PSN were found to deliver the −20 °C isotherm deeper and more quickly.

As important as the speed in reaching freezing temperatures is the prompt cessation of freezing upon completion of the ablative procedure. This is essential to prevent inadvertent damage to tissue beyond the intended target area. This can often be an issue encountered with other ablative methods like radiofrequency (RF) ablation and radiation, wherein after removal of the energy source, a substantial reservoir of energy at the center of the lesion persists and continues to radiate outward, resulting in “lesion growth” post-treatment [27,29,43]. For the treatment of PDAC, continued lesion growth can result in unintended damage to pancreatic tissue, blood vessels, stomach wall, etc., depending on tumor location. Cryoablation, as an energy-extractive treatment, allows heat energy from the surrounding tissue to flow into the lesion once the freeze procedure is completed. This mechanism helps prevent further spread and significantly decreases the risk of damage to surrounding tissues and organs. To this end, the data demonstrated that upon stopping cryogen flow, temperatures began to warm nearly instantaneously (within 5 s) for the *FrostBite* EUS cryocatheter. (Figure 2).

Beyond freeze performance in ultrasound gel, we assessed the ablative potential of *FrostBite* under physiological heat loads using a TEM. By combining the capacity to evaluate freezing characteristics and cellular responses, the TEM model has proven to be a versatile tool in assessing cellular responses to cryosurgery and cryosurgical technologies. This model has been utilized in the examination of treatments for various tissues, including prostate [69,72], kidney [85], and bladder [86] tissues. Employing this test setup, a transesophageal cryoablative procedure of PDAC can be modeled ex vivo using tissue geometries and heat loads similar to physiological conditions. Accordingly, TEMs were frozen using a single 5 min freeze protocol, iceball size was measured following the procedure, and the resultant ablative (necrotic) zone was assessed 1 and 3 days post-treatment. Fluorescence imaging 1 day post-thaw revealed a necrotic diameter of 20.1 mm (±2.1) compared to an overall freeze zone diameter of 25.1 mm (±0.4). This equated to ~54% of the total frozen volume being ablated following a single 5 min freeze (6.9 cm^3^ vs. 13 cm^3^), indicating a marginal zone of incomplete cell death of <~2.5 mm (Table 4). By 72 h post-thaw, the ablative volume had decreased slightly to 6.8 cm^3^ (*p* = 0.14) (Table 4). Studies have shown that this is due to cellular recovery, infiltration, and regrowth in the outer region of the cryogenic lesion where non-lethal temperatures are experienced (>−20 °C to 37 °C) [67,69,72]. Clinically, this is addressed by the application of a positive freeze margin to ensure that the entire tumor is exposed to lethal temperatures [24,25,30,31]. Despite the observed in vitro-specific recovery, the region of incomplete cell death was still less than 2 mm at 72 h post-thaw, suggesting that only a 2 mm positive freeze margin is necessary with *FrostBite*. This is significantly smaller than the ≥5 mm positive freeze margin typically applied when using today’s argon-based JT cryodevices [24,69,83,84,87]. Correlation of isotherm and 72 h viability data demonstrated complete PANC-1 cell death in the range of −20 °C to −25 °C following a single 5 min freeze using the *FrostBite* EUS cryocatheter. This corroborates previous in vitro studies, which have reported −25 °C to be the minimal critical temperature for PDAC cells (PANC-1 and BxPC-3) [68,70,71].

Having demonstrated *FrostBite*’s ability to rapidly and precisely deliver lethal temperatures nearer to the periphery of the iceball using PSN, we conducted a preliminary porcine study to assess its performance. In vivo studies revealed the consistent creation of a 2.03 cm × 3.2 cm (dia × length) cryolesion, on average, following a single 3 min freeze protocol (Table 5). Extension of the freezing to 5 min resulted in an increase in lesion size to 2.2 cm × 3.4 cm. The application of a repeat freeze (double 3 min freeze (3/3/3)) yielded a further increase to 2.5 cm v 3.5 cm. Comparison of the pathological lesion measurements with measurements of lesion diameter taken via ultrasound at the end of the freeze intervals revealed a high degree of correlation between the assessment modalities. For example, measurement of iceball size via EUS at the end of a single 3 min freeze yielded an average diameter of 2.01 cm (±0.04), which matched physical pathological measurement post-sacrifice at an average lesion diameter of 2.03 cm (±0.05) following a single 3 min freeze. Similar results were seen with the single 5 min freeze (EUS: 2.1 cm vs. Path: 2.2 cm) and repeat 3 min freeze (EUS 2nd freeze: 2.4 cm vs. Path: 2.5 cm) (Table 5). In addition to the overall size and consistency of lesion formation, we also assessed the impact of freezing with *FrostBite* on targets in close proximity to vasculature. When the ablation zone of *FrostBite* was positioned ~0.5 cm from a blood vessel and a single 3 min freeze was applied, the cryolesion was found to grow around/sculpt to the blood vessel. This sculpting was observed both in real time under ultrasound imaging and upon pathological examination (Figure 5G and Figure 6D). This is hypothesized to be a result of the blood flow within the vessel providing a protective insulating heat source preventing freezing of the blood vessel. Doppler monitoring of blood flow within the vessel throughout the freeze procedure revealed no apparent impact of the freezing on blood flow during or following freezing (Figure 5F,H). The ability to freeze targets in close proximity to blood vessels is important, as vasculature involvement is a contraindication for surgical removal as well as hyperthermic (heat-based) ablation. Heat damage has been shown to radiate into and be amplified by blood vessels, resulting in extensive damage to the involved vasculature. As such, the ability to cryoablate lesions with vasculature involvement using *FrostBite* could be highly beneficial for patients for whom ablation treatment is not an option. While a pilot study, these results nonetheless suggest that *FrostBite* may provide a new treatment strategy for treating PDAC. Additional studies are necessary to fully answer this question.

Previous investigations into the cryoablation of PDAC have focused on using argon-JT and percutaneous needles inserted through the abdomen [4,5,42,60,61,62,63,64]. While this approach has shown promise, due to the physical location of the pancreas, the use of percutaneous cryoablation is restricted to the tail of the pancreas, thereby highly limiting its use in treating PDAC. Further, current percutaneous argon JT-based cryoneedles also require long freeze times (e.g., 10 min dual freezes) to create an effective cryolesion [69,83,84]. Additionally, these percutaneous argon cryoprobes have been shown to have a 5 mm to 7.5 mm zone of partial damage (0 °C to −20 °C) associated with the iceball margin [69,72,83,84]. As such, a high level of damage to non-targeted tissue (collateral damage) is associated with argon cryoprobes. In pancreatic applications, this may increase the risk of developing pancreatitis following a procedure. Given the acute, feasible nature of this study coupled with the in vivo cryoablations being performed on liver tissue, a specific investigation into the incidence of pancreatitis post-treatment was not conducted. While not investigated herein, several clinical studies have reported on the use of percutaneous cryoablation to treat pancreatic cancer (Table 6) [5,61,88,89,90]. While limited, analysis of these clinical studies suggests a ~6% overall incidence of pancreatitis post-cryoablation. (9/159 patients; 8 acute and 1 severe). Additionally, pre-clinical porcine studies also suggest a low incidence of pancreatitis post-cryo treatment [4]. As the development of pancreatitis post-treatment is a critical consideration in treatment safety and efficacy, future chronic studies will include analyses of the development of pancreatitis. We hypothesize that the ability to deliver a colder, more precise cryoablative insult via an EUS approach in less time should have numerous benefits over current hyperthermia (RFA, HiFu, PFA, etc.) and percutaneous cryoablative technologies.

While the results of this study are promising, the in vivo portion has several limitations. Firstly, this study was limited to eight lesions that were created in healthy liver tissue. This prevented analysis of the incidence of pancreatitis post-treatment. Secondly, analysis was limited to gross pathology in an acute porcine model. This prevented detailed histological characterization of the ablated tissue areas. Thirdly, while studies have shown that liver and pancreatic tissue have a similar minimal lethal temperature range (−20 °C to −25 °C) [26,68,70,71,91], extrapolation of the in vivo findings on liver tissue cryoablation to that of pancreatic tissue cryoablation using *FrostBite* should be with caution. To address these items, future in vivo studies will include expanded survival studies, histological analysis (H&E and trichrome staining), in vivo tumor models, and investigation of freeze duration (time and repeat freezes) on lesion size. This will effectively establish “dosing” parameters that result in tumor destruction while eliminating the need for over-freezing, thereby reducing damage to non-targeted tissue and vasculature. Importantly, future studies will include analyses of the incidence of pancreatitis post-cryoablation with *FrostBite*. Once completed, future clinical studies will be conducted to validate the safety, efficacy, post-procedural incidence of pancreatitis, and long-term outcomes in pancreatic cancer patients.

## 5. Conclusions

Given the recent success and growth of cryoablation, there is a growing trend toward increased utilization and demand for enhanced procedural outcomes. To achieve this efficiently, new cryotechnologies are being developed to facilitate rapid tissue cooling for more effective tissue ablation. We hypothesized that using *FrostBite,* rapid and effective ablation of tissue could be achieved via a minimally invasive endoscopic ultrasound-based approach. Through a series of laboratory and pre-clinical evaluations, in this study, we established that the *FrostBite* EUS cryocatheter was able to deliver rapid and effective ablative doses in a controlled and predictable manner. The data demonstrate that *FrostBite* achieves swift and efficient delivery of a cryoablative “thermal dose” (−20 °C) deep into a frozen tissue mass within a brief 3 min timeframe under physiological conditions (37 °C). In vivo studies demonstrated that cryolesion formation was successfully visualized in real time using ultrasound. Importantly, effective ablation was attained next to vasculature without damaging the blood vessel. This may have implications for targeting unresectable PDAC. Although the findings of this study are promising, their investigational nature imposes limitations on the extent of conclusive statements. Nevertheless, the data strongly support the potential of this technology, warranting further in vivo assessments. In summary, the results indicate that the *FrostBite* EUS cryocatheter shows promise as a next-generation cryoablation device, enabling swift and controlled application of ultra-cold temperatures for efficient freezing of targeted tissue, including PDAC, via an endoscopic approach.

## Figures and Tables

**Figure 1 biomedicines-12-00507-f001:**
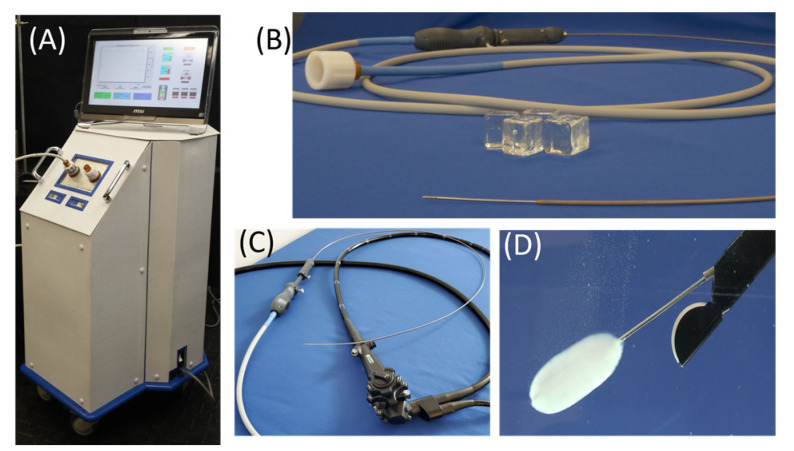
Photographs of the *FrostBite* EUS cryocatheter and PSN cryoconsole. (**A**) Image of the PSN cryoconsole. (**B**) Image of *FrostBite* with the distal cryoneedle extended. (**C**) Image of *FrostBite* in relation to an EUS scope. (**D**) Image of the ice ball created by the *FrostBite* cryoneedle during a 3 min freeze in water when inserted through the working channel of an EUS scope.

**Figure 2 biomedicines-12-00507-f002:**
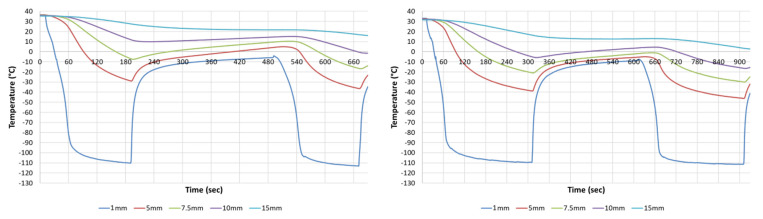
Real-time temperature profiles generated by the *FrostBite* EUS cryocatheter during various freezing protocols. Temperatures at the middle of the freeze zone were monitored during a repeat 3 and 5 min freeze procedure. Temperatures were found to rapidly drop next to the cryoprobe surface while decreasing more gradually as the distance from the surface increased. Analysis revealed the −20 °C isotherm to reach a diameter of ~1.2 cm following a single 3 min freeze, whereas a single 5 min freeze yielded a −20 °C diameter of ~1.5 cm. Application of a repeat 3 and 5 min freeze protocol resulted in an increase of the −20 °C isotherm to 1.4 cm and 1.9 cm, respectively.

**Figure 3 biomedicines-12-00507-f003:**
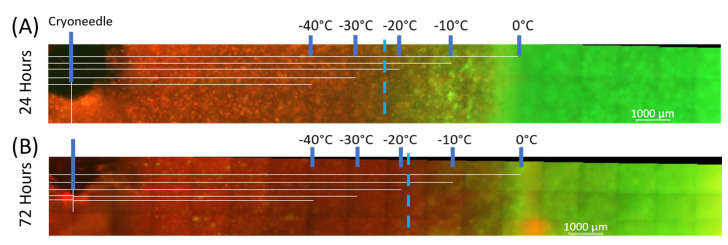
Fluorescent micrographs of the PDAC-TEMs following single 5 min freeze. At 24 h (**A**) and 72 h (**B**) post-freeze, the extent of PANC-1 cell death in TEMs was evaluated via fluorescence microscopy following staining with Calcein-AM (green, live) and propidium iodide (red, dead). Using the Zeiss ZEN software, measurements were made, and then temperatures from the corresponding thermocouples were overlayed. Analysis revealed complete PANC-1 cell destruction was attained following exposure to a temperature of ~−20 °C or colder.

**Figure 4 biomedicines-12-00507-f004:**
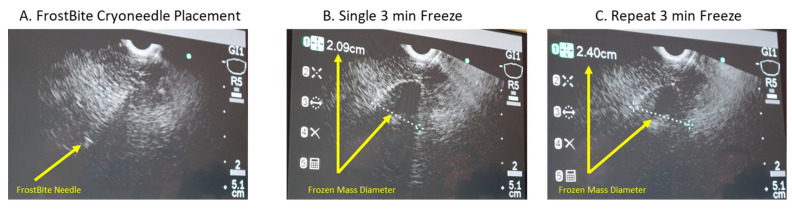
Endoscopic ultrasound images of a freeze procedure. (**A**) Visualization of the *FrostBite* cryoneedle inserted into liver tissue prior to freeze initiation. (**B**,**C**) Visualization and measurement of the resultant frozen tissue masses following a (**B**) single 3 min and (**C**) repeat 3 min freeze. Measurement (dotted line) of the frozen tissue mass diameter via endoscopic ultrasound revealed the single freeze protocol (**B**) yielded a 2.09 cm diameter frozen mass, whereas the repeat 3 min freeze protocol (3/3/3) (**C**) yielded a 2.4 cm diameter frozen tissue mass.

**Figure 5 biomedicines-12-00507-f005:**
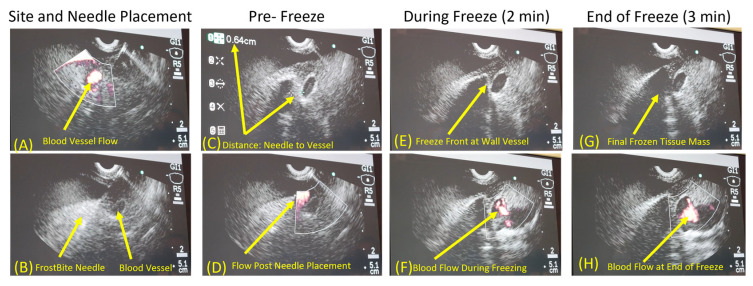
Endoscopic ultrasound Doppler imaging of a single 3-minute freeze protocol next to a blood vessel. Images demonstrate blood flow through the vessel prior to *FrostBite* placement (**A**), visualization of *FrostBite* needle placement (**B**), measurement of the distance between the blood vessel and *FrostBite* (**C**), and Doppler blood flow prior to freeze initiation (**D**). Lesion visualization of iceball approaching the blood vessel (**E**) and blood flow in vessel (**F**) following 2 min of freezing. Iceball growing around blood vessel (**G**) and Doppler assessment of blood flow (white region) in vessel (**H**) at the end of the 3 min freeze protocol. Images show that when *FrostBite* was placed ~0.6 cm from a blood vessel and a 3 min freeze protocol was performed, the frozen tissue mass sculpted around the blood vessel, and the freezing process did not impact blood flow through the vessel throughout the freeze procedure.

**Figure 6 biomedicines-12-00507-f006:**
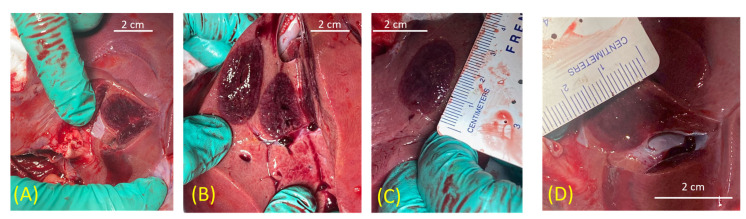
Gross pathology of a resected porcine liver tissue following freezing. Following an EUS-based cryoablation procedure using *FrostBite*, porcine livers were resected and analyzed. Gross pathology of the tissue revealed the consistent creation of ~2 cm × 3.2 cm lesions (deep purple tissue) following a single 3 min freeze (**A**,**B** (right lesion) and **C**). A double 3 min freeze resulted in an increase in the lesion size to ~2.5 cm × 3.5 cm (B, left lesion). Analysis of a freeze lesion created near a blood vessel (**D**) confirmed the EUS imaging of the cryolesion sculpting around the blood vessel, and the vessel wall showed no obvious signs of damage.

**Table 1 biomedicines-12-00507-t001:** Iceball Dimensions Following Various Freeze Protocols in 35 °C Acellular Hydrogel.

Freeze Protocol (mins)	Time to Ice (sec)	Iceball Size (mm (±SD))	
		Dia.	Length
Single 3	18.1 (±1.9)	17.1 (±0.2)	32.4 (±1.0)
Single 5	17.8 (±1.6)	21.7 (±0.8)	35.1 (±1.7)
Double 3/5/3	15.7 (±1.5)	20.6 (±0.5)	34.2 (±0.3)
Double 5/5/5	16.0 (±1.6)	27.5 (±1.1)	38.2 (±2.1)

**Table 2 biomedicines-12-00507-t002:** Isotherm Diameters for the FrostBite EUS-Cryocatheter Following Freezing in a 35 °C Acellular Hydrogel.

Freeze Protocol (mins)	Single (First) Freeze Isotherm Dia (mm(±SD))					Repeat (Second) Freeze Isotherm Dia (mm(±SD))				
	0 °C	−10 °C	−20 °C	−30 °C	−40 °C	0 °C	−10 °C	−20 °C	−30 °C	−40 °C
Single 3	16.4 (0.5)	13.9 (0.4)	11.6 (0.5)	9.6 (0.3)	8.5 (0.3)					
Single 5	21.2 (0.7)	17.8 (0.4)	14.8 (0.4)	12.1 (0.4)	9.7 (0.2)					
Double 3/5/3	16.8 (0.6)	14.2 (0.5)	12.0 (0.5)	9.8 (0.4)	8.6 (0.2)	20.0 (0.4)	17.0 (0.7)	14.0 (0.6)	11.6 (0.5)	9.6 (0.4)
Double 5/5/5	22.2 (0.7)	18.4 (0.6)	15.2 (0.4)	12.4 (0.5)	9.8 (0.2)	28.2 (1.0)	23.2 (0.9)	18.6 (0.6)	15.0 (0.6)	11.8 (0.4)

**Table 3 biomedicines-12-00507-t003:** Cooling Power Assessment for the FrostBite EUS Cryocatheter Following a 3 to 5 min Freeze Protocol.

Time	Water Temperature (°C)			Cooling Power (W)	
	Start Temp	End Temp	ΔT	Total Watts (W)	W/mm^2^
3 min	33.6 (±0.3)	28.7 (±0.4)	4.9 (±0.8)	51.2 (±6.3)	1.09 (±0.04)
3.5 min		27.9 (±0.4)	5.7 (±0.9)	51.6 (±5.8)	1.09 (±0.03)
4 min		27.0 (±0.5)	6.6 (±1.0)	52.0 (±5.9)	1.1 (±0.05)
4.5min		26.1 (±0.5)	7.4 (±1.1)	52.2 (±5.5)	1.11 (±0.04)
5 min		25.1 (±0.6)	8.4 (±1.2)	53.3 (±5.3)	1.13 (±0.2)
Avg	33.6 (±0.3)	27.0 (±0.5)	6.6 (±1.0)	52.1 (±5.7)	1.1 (±0.03)

**Table 4 biomedicines-12-00507-t004:** Diameters and Areas (±SD) of Frozen and Ablative Zones for PANC-1 TEMs at 24 h and 72 h Following Single 5 min Freeze Protocol.

		Single 5 min Freeze			
		24 h Post-Thaw		72 h Post-Thaw	
		Freeze Zone	Necrotic Zone	Freeze Zone	Necrotic Zone
**PANC-1 TEMs**	**Dia.** **(mm (±SD))**	25.1 (0.4)	20.1 (2.1)	24.1 (2.0)	20.6 (1.4)
	**Volume (cm^3^ (±SD))**	13.0 (0.4)	6.9 (0.8)	12.1 (1.9)	6.8 (0.6)
	**% Lethality**		53.3		56.1

**Table 5 biomedicines-12-00507-t005:** Analysis of Cryolesions Created by FrostBite in Porcine Liver Tissue.

Freeze #		1	2	3	4	5	6	7	8
Freeze Protocol		Single	Single	Single	Single	Single	Single	Double	Single
Freeze Duration		3 min	3 min	3 min	5 min	3 min	3 min	3/3/3	3 min
**Frozen Mass Size**	**EUS Diameter**	N/A	2 cm	2 cm	2.1 cm	1.9 cm	N/A	1st freeze: 2.1 cm 2nd freeze: 2.4 cm	N/A
	**Pathology Dia × Lgn**	2.1 cm × 3.2 cm	2.1 cm × 3.3 cm	2 cm × 3.1 cm	2.2 cm × 3.4 cm	2 cm × 3 cm	N/A *	2.5 cm × 3.5 cm	N/A *

* Lesions 6 and 8 were found to overlap upon gross pathology analysis so measurement of lesion size was not recorded.

**Table 6 biomedicines-12-00507-t006:** Overview of Clinical Pancreatic Cancer Cryoablation Literature.

				% of Patients	
Author	# Patients	Average Survival (months)	% Survival @ 12 months	Pain Reduction	Incidence of Pancreatitis
Kovach et al. [5]	9	5	-	66%	0% (0/9)
Niu et al. [88]	32	15.9	54%	>50%	0% (0/32)
Xu et al. [60]	49	16	63%	-	12% (6/49)
Xu et al. [61]	59	8.4	34%	-	5% (3/59)
Wu et al. [90]	10	3 month follow-up	-	6.9 to 2.0	0% (0/10)

## Data Availability

The data that support the findings of this study are available from CPSI Biotech, but restrictions apply to the availability of these data, which were used under license for the current study and so are not publicly available. Data are, however, available from the authors upon reasonable request and with permission of CPSI Biotech.

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
