# Peer review of "The Assessment of a Novel Endoscopic Ultrasound-Compatible Cryocatheter to Ablate Pancreatic Cancer"

_biomedicines, 2024, doi:10.3390/biomedicines12030507_

Round 1

Reviewer 1 Report

Comments and Suggestions for Authors

In this study, live pigs were treated with a new cryoablation catheter using in vivo methodology. Unlike the currently deployed cryoablation methods in the esophagus and stomach, this new catheter avoids volume expansion of liquid nitrogen, making it suitable for closed tissue spaces. This study is believed to be the first porcine study using this novel catheter, holding significant potential for downstream medical applications.

The experimental design and post-ablation evaluation were conducted meticulously and in detail. The investigative team covered multiple aspects in terms of technical knowledge and information. However, as a physician/surgeon/procedure expert, I have some key questions that the authors need to address:

  1. The ablation was performed in live pig models under anesthesia. Did the authors check blood levels of amylase and lipase post-ablation to determine if the pigs had post-procedure pancreatitis? While I understand the pigs were sacrificed for the experiment, post-procedure pancreatitis is a crucial adverse event in all pancreas-related ablation procedures. There is no mention of pancreatitis in the entire study, and as a provider involved in performing multiple pancreas procedures, post-procedure enzyme measurements (in the absence of symptoms) would be an essential outcome to consider.
  2. Are there any histopathological images for review, such as the ablation zone, peri-ablation zone, or other relevant areas? Did a pathologist review these for changes indicative of pancreatitis? How severe were the changes of pancreatitis in the peri-ablation zone?
  3. While achieving successful lesion ablation is the treatment goal for tumors, especially in procedures like microwave ablation in surgical lesions of the liver, the pancreas is particularly sensitive to post-procedure pancreatitis. Since this experiment focuses on the pancreas, the authors are strongly encouraged to provide data on points #1 and #2 and discuss them accordingly.

Finally, I congratulate the investigators on developing this novel probe with a strong potential for future applications in managing precancerous and malignant tissue.

Author Response

Reviewer: 1

Initial Comment to the Author:  In this study, live pigs were treated with a new cryoablation catheter using in vivo methodology. Unlike the currently deployed cryoablation methods in the esophagus and stomach, this new catheter avoids volume expansion of liquid nitrogen, making it suitable for closed tissue spaces. This study is believed to be the first porcine study using this novel catheter, holding significant potential for downstream medical applications.

The experimental design and post-ablation evaluation were conducted meticulously and in detail. The investigative team covered multiple aspects in terms of technical knowledge and information. However, as a physician/surgeon/procedure expert, I have some key questions that the authors need to address:

Response: We thank the reviewer for the positive feedback and comments. We agree that while much work remains, this technology has significant downstream clinical potential.  

Specific Comments:
Comment 1: The ablation was performed in live pig models under anesthesia. Did the authors check blood levels of amylase and lipase post-ablation to determine if the pigs had post-procedure pancreatitis? While I understand the pigs were sacrificed for the experiment, post-procedure pancreatitis is a crucial adverse event in all pancreas-related ablation procedures. There is no mention of pancreatitis in the entire study, and as a provider involved in performing multiple pancreas procedures, post-procedure enzyme measurements (in the absence of symptoms) would be an essential outcome to consider.

Response: Thank you for the comment. As the in vivo portion of the study was designed as an acute feasibility study with ablation in the liver we were not able to conduct any analysis of pancreatitis following treatment. We agree that an understanding of the incidence of pancreatitis following cryoablation is an important factor. As we recognize the importances of this topic, the revised manuscript now includes expanded discussion, clinical literature data summary (Table 6) and corresponding reference on the incidence of pancreatitis following cryoablation. The published clinical literature suggests an overall ~6% incidence of acute pancreatitis post cryoablation. As studies have yet to be conducted specifically using FrostBite, we have also added discussion of this in the study limitations and have indicated the need for investigation in future chronic studies.

Comment 2: Are there any histopathological images for review, such as the ablation zone, peri-ablation zone, or other relevant areas? Did a pathologist review these for changes indicative of pancreatitis? How severe were the changes of pancreatitis in the peri-ablation zone?

Response: As the in vivo portion of the study was acute no histopathological analysis was conducted. In depth histological analysis of acute cryolesions typically does not provide any additional information compared to gross pathological analysis.  Further as the in vivo lesion were performed in the liver, the changes in the tissue would not provide insight into potential pancreatitis.  We have identified this as a limitation and the need for continued study.

Comment 3: While achieving successful lesion ablation is the treatment goal for tumors, especially in procedures like microwave ablation in surgical lesions of the liver, the pancreas is particularly sensitive to post-procedure pancreatitis. Since this experiment focuses on the pancreas, the authors are strongly encouraged to provide data on points #1 and #2 and discuss them accordingly.

Response: We agree that the incidence of pancreatitis is an important factor. While it was not analyzed in the current study, we have added discussion of this to the revised manuscript as well as identified this as a limitation and the need for further study.

Reviewer 2 Report

Comments and Suggestions for Authors

Thanks to the authors for sharing a significant study of EUS-guided cryoablation treatment for PDAC. This study has a novel design, rich data, and high potential for future clinical application. I fully support the publication of this manuscript and offer the following suggestions.

1. It is recommended that some pictures of the cryoablation system and EUS-compatible cryocatheter be added.

2. Figure 3 and 4 appear to be photos taken from the display of the endoscopic equipment, and these images were all tilted. It is recommended to use high-quality images retained by the EUS devices and upload a separate file for each EUS image to improve the readers' reading experience and the promotion value of this manuscript.

3. Given that the topic of this manuscript is the role of new technology for PDAC cryoablation, but porcine liver was used instead of pancreas, it is suggested to include some limitations in the Discussion, such as the possible differences between liver and pancreas cryoablation and the inability to monitor postoperative pancreatitis.

Author Response

Reviewer: 2

Initial Comments to the Author: Thanks to the authors for sharing a significant study of EUS-guided cryoablation treatment for PDAC. This study has a novel design, rich data, and high potential for future clinical application. I fully support the publication of this manuscript and offer the following suggestions.

Response: We thank the reviewer for their complimentary and encouraging comments.  We appreciate the feedback and suggestions.

Specific Comments:
Comment 1:  It is recommended that some pictures of the cryoablation system and EUS-compatible cryocatheter be added.

Response: A new figure with pictures of FrostBite and the PSN cryodevice has been added to the manuscript (new fig, Fig. 1).

Comment 2: Figure 3 and 4 appear to be photos taken from the display of the endoscopic equipment, and these images were all tilted. It is recommended to use high-quality images retained by the EUS devices and upload a separate file for each EUS image to improve the readers' reading experience and the promotion value of this manuscript.

Response: Thank you for the comment. The reviewer is correct the EUS images were captured from the display. We agree that untitled images could enhance reading experience promotional value, unfortunately only images and video captured from the EUS monitor display are available.

Comment 3: Given that the topic of this manuscript is the role of new technology for PDAC cryoablation, but porcine liver was used instead of pancreas, it is suggested to include some limitations in the Discussion, such as the possible differences between liver and pancreas cryoablation and the inability to monitor postoperative pancreatitis.

Response: As recommended, the revised manuscript now included discussion comparing liver and pancreatic cancer response to cryoablation (including additional citations).  We have also added discussion on the limitation of not being able to assess pancreatitis due to both ablating liver tissue and the acute nature of the animal study. To further address this important topic, the revised manuscript now includes expanded discussion, clinical literature data summary (Table 6) and corresponding reference on the incidence of pancreatitis following cryoablation. The published clinical literature suggests an overall ~6% incidence of acute pancreatitis post cryoablation.

Reviewer 3 Report

Comments and Suggestions for Authors

1.      The study introduces the FrostBite cryocatheter as a potential advancement in the minimally invasive treatment of pancreatic cancer.

2.      While the results from gel models, tissue-engineered models, and porcine studies are promising, the lack of human trials and long-term data necessitates caution.

3.      The cryocatheter's compatibility with endoscopic ultrasound and its ability to achieve controllable freezing represent encouraging features.

4.      Future research should focus on clinical trials to validate the safety, efficacy, and long-term outcomes of FrostBite in human subjects.

Author Response

Reviewer 3

Specific Comments

Comment 1: The study introduces the FrostBite cryocatheter as a potential advancement in the minimally invasive treatment of pancreatic cancer.

Response: We thank the reviewer for the comment and recognition of the potential for FrostBite.  While we recognize much work remains, we believe there is tremendous potential for this technology.

Comment 2: While the results from gel models, tissue-engineered models, and porcine studies are promising, the lack of human trials and long-term data necessitates caution.

Response: We completely agree.  We have added discussion of the limitations of the current study as well as have identified the need for continued study to address this important point.  We have also revised the text to caution the reader in interpretation of study findings to potential clinical outcome as recommended.

Comment 3: The cryocatheter's compatibility with endoscopic ultrasound and its ability to achieve controllable freezing represent encouraging features.

Response: We thank the reviewer for this comment and observation.  We agree, EUS compatibility and the precision and repeatability of the lesion created with FrostBite is encouraging.

Comment 4: Future research should focus on clinical trials to validate the safety, efficacy, and long-term outcomes of FrostBite in human subjects.

Response: We completely agree. We have added discussion on the need for continued study to address these important points.